

# MX1 and UBE2L6 are potential metaflammation gene targets in both diabetes and atherosclerosis

Guisheng Wang[1],[*], Rongrong Hua[1],[*], Xiaoxia Chen[1], Xucheng He[1], Yao Dingming[1], Hua Chen[2], Buhuan Zhang[1], Yuru Dong[1], Muqing Liu[1], Jiaxiong Liu[1], Ting Liu[3], Jingwei Zhao[4], Yu Qiong Zhao[2] and Li Qiao[5]

[1] Department of Radiology, Third Medical Center of Chinese PLA General Hospital, Beijing, China
[2] Laboratory Animal Center, Chinese People's Liberation Army General Hospital, Beijing, China
[3] Department of Radiology, Beijing YouAn Hospital, Capital Medical University, Beijing, China
[4] Department of Radiology, National Cancer Center/National Clinical Research Center for Cancer/Cancer Hospital, Chinese Academy of Medical Sciences and Peking Union Medical College, Beijing, China
[5] Department of International Business, Business College of Beijing Union University, Beijing, China

[*] These authors contributed equally to this work.

Corresponding authors
Yu Qiong Zhao,
zhaoyuqiong@163.com
Li Qiao, li.qiao@buu.edu.cn

## ABSTRACT

**Background:** The coexistence of diabetes mellitus (DM) and atherosclerosis (AS) is widespread, although the explicit metabolism and metabolism-associated molecular patterns (MAMPs) responsible for the correlation are still unclear.

**Methods:** Twenty-four genetically wild-type male Ba-Ma mini pigs were randomly divided into five groups distinguished by different combinations of 90 mg/kg streptozotocin (STZ) intravenous injection and high-cholesterol/lipid (HC) or high-lipid (HL) diet feeding for 9 months in total. Pigs in the STZ+HC and STZ+HL groups were injected with STZ first and then fed the HC or HL diet for 9 months. In contrast, pigs in the HC+STZ and HL+STZ groups were fed the HC or HL diet for 9 months and injected with STZ at 3 months. The controls were only fed a regular diet for 9 months. The blood glucose and abdominal aortic plaque observed through oil red O staining were used as evaluation indicators for successful modelling of DM and AS. A microarray gene expression analysis of all subjects was performed.

**Results:** Atherosclerotic lesions were observed only in the HC+STZ and STZ+HC groups. A total of 103 differentially expressed genes (DEGs) were identified as common between them. The most significantly enriched pathways of 103 common DEGs were influenza A, hepatitis C, and measles. The global and internal protein–protein interaction (PPI) networks of the 103 common DEGs consisted of 648 and 14 nodes, respectively. The top 10 hub proteins, namely, ISG15, IRG6, IRF7, IFIT3, MX1, UBE2L6, DDX58, IFIT2, USP18, and IFI44L, drive aspects of DM and AS. MX1 and UBE2L6 were the intersection of internal and global PPI networks. The expression of MX1 and UBE2L6 was 507.22 ± 342.56 and 96.99 ± 49.92 in the HC+STZ group, respectively, which was significantly higher than others and may be linked to the severity of hyperglycaemia-related atherosclerosis. Further PPI network analysis of calcium/micronutrients, including MX1 and UBE2L6, consisted of 58 and 18 nodes, respectively. The most significantly enriched KEGG pathways were
glutathione metabolism, pyrimidine metabolism, purine metabolism, and metabolic pathways.

**Conclusions:** The global and internal PPI network of the 103 common DEGs consisted of 648 and 14 nodes, respectively. The intersection of the nodes of internal and global PPI networks was MX1 and UBE2L6, suggesting their key role in the comorbidity mechanism of DM and AS. This inference was partly verified by the overexpression of MX1 and UBE2L6 in the HC+STZ group but not others. Further calcium- and micronutrient-related enriched KEGG pathway analysis supported that MX1 and UBE2L6 may affect the inflammatory response through micronutrient metabolic pathways, conceptually named metaflammation. Collectively, MX1 and UBE2L6 may be potential common biomarkers for DM and AS that may reveal metaflammatory aspects of the pathological process, although proper validation is still needed to determine their contribution to the detailed mechanism.

# INTRODUCTION

Diabetes mellitus (DM) involves hyperglycaemia due to insulin inactivity or insufficiency and is increasingly becoming a global health issue due to its high morbidity and mortality. A growing body of evidence has established that DM patients have an increased risk of developing atherosclerosis (AS). The higher concurrency rate and severity of AS in DM patients with numerous cardiovascular risk factors may result from their metabolic abnormalities (*Beckman, Creager & Libby, 2002*; *Yuan et al., 2017*). The apparent association characterized by the coexistence of DM and AS seems to be widely recognized. However, in a meta-analysis of 17,330 participants for periodontal disease and carotid atherosclerosis, subgroup analysis of adjusted DM showed only borderline significance (OR: 1.08; 95% CI [1.00–1.18]; $p = 0.05$) (*Whayne, 2017*), making the relationship even more confusing.

Simultaneously, there is convincing evidence for pathological cross connections between DM and AS (*Poznyak et al., 2020*). First, an imbalance in cholesterol homeostasis is involved in the progression of cardiovascular disease (CVD) related to DM. Previously, the ratio of triglycerides (TGs) to high-density lipoprotein cholesterol (HDL-C) was reported to predict the incidence of type 2 diabetes (T2D) within 10 years (*Yuge et al., 2023*). In patients with T2D, remnant cholesterol rather than others was an independent risk factor for CVD (*Huh et al., 2022*). More in-depth research revealed that insulin reduces 12α-hydroxylated bile acids, cholesterol absorption, and plasma cholesterol levels by inhibiting FoxO1, which may contribute to the high risk of CVD in diabetes (*Semova et al., 2022*). Second, micronutrient and metabolic imbalances are related to the occurrence and progression of DM and AS. Accumulating evidence has indicated that impaired intracellular calcium homeostasis (*Tammineni et al., 2020*; *Zhang et al., 2023*) and zinc (*Tamura, 2021*; *Wang, Huang & Yang, 2023*) or selenium (*Dabravolski et al., 2023*;

*Hamdan, Hamdan & Adam, 2022*; *Steinbrenner, Duntas & Rayman, 2022*) deficiency promote hyperglycaemia and diabetes. Numerous studies have also shown that zinc (*Tamura, 2021*; *Wang, Huang & Yang, 2023*) or selenium (*Dabravolski et al., 2023*; *Hamdan, Hamdan & Adam, 2022*; *Steinbrenner, Duntas & Rayman, 2022*) deficiency may be associated with risk factors for DM and AS. Finally, and most importantly, inflammation, metaflammation, and immunometabolic disorders indicate the imbalance between the immune response and metabolism, having an essential influence on many pathological states, such as DM, AS and other chronic noncommunicable diseases (*Ahmed et al., 2018*; *Hotamisligil, 2017*; *Ma et al., 2017*; *Olcay et al., 2019*; *Rao et al., 2022*). Metaflammation, as the concept explaining the coincidence of DM and AS (*Christ & Latz, 2019*), is a long-term chronic low-grade inflammation status caused by metabolic factors, with a similar local inflammatory response and signalling pathways to conventional inflammation (*Hotamisligil, 2006*; *Kurylowicz & Kozniewski, 2020*). Unfortunately, there are no explicit metabolism-associated molecular patterns (MAMPs) (*Wang et al., 2020*) or pathways responsible for the correlation between these two conditions, although hyperglycaemia and free fatty acids are considered important types of MAMPs that can induce insulin resistance through NOD-like receptors (*Koenen et al., 2011*) or toll-like receptor signalling (*Rada et al., 2018*), respectively.

In our previous study, T2D models of Ba-Ma mini pigs were established by feeding an experimental diet of high-lipid (HL) or high-cholesterol/lipid (HC) and then administering 90 mg/kg streptozotocin (STZ) by intravenous injection. As a naturally occurring alkylating antineoplastic agent, STZ is particularly toxic to insulin-producing beta cells in the mammalian pancreas, and can be used alone or in combination with high-fat diet to induce models of diabetes. Unexpectedly, plaque lesions of the abdominal aorta in the HC groups were more obvious than those in the HL groups, suggesting that the pathological mechanisms of DM and AS overlap to some extent rather than completely. Therefore, transcriptome experiments and bioinformatics analysis were used to search for differentially expressed genes and significantly enriched pathways common in DM and AS, with the aim of discovering the potential connected targets.

## MATERIALS AND METHODS

### Animal models and interventions

The animal study protocol was approved by the Ethics Committee of the Third Medical Center of Chinese PLA General Hospital (KY2021-008; 2021/02/02). Pigs have been widely used as models in studies on metabolism and cardiovascular disease because they share similarities with humans in anatomy, physiology, and genetics (*Zhao et al., 2018*). As described previously (*Zhao et al., 2022*), 24 genetically wild-type male Ba-Ma mini pigs (5–6 months, 12.6–15.2 kg) were purchased from Beijing Strong Century Minipigs Breeding Base and randomly divided into five groups. Pigs in the STZ+HC and STZ+HL groups were injected with STZ first and then fed the HC or HL diet for 9 months. In contrast, pigs in the HC+STZ and HL+STZ groups were fed the HC or HL diet for 9 months and injected with STZ at 3 months. The control group was only fed a regular diet for 9 months. The proportions of the main components of tallow/cholesterol/bile salt/basic

feed were 15/0/0.5/84.5% and 15/2/0.5/82.5% in the HC and HL experimental diets, respectively. The animals were housed in separate cages (20–26 °C, relative humidity of 40–70%, and a light/dark cycle of 12/12 h). After deletion of the missing data, 20 cases were included in the study: control group ($n = 4$), HC+STZ group ($n = 4$), STZ+HC group ($n = 3$), HL+STZ group ($n = 4$), and STZ+HL group ($n = 5$). Anaesthesia for all animals was induced by intramuscular injection of xylazine hydrochloride (25 mg/kg) and midazolam (0.25 mg/kg) and maintained by intravenous injection of pentobarbital sodium (6 mg/kg), after which the femoral artery was bled until the animals died. The blood glucose levels and the size of abdominal aorta atherosclerotic plaque are the basis to judge the success of DM and AS modelling. The plaques and their sizes were observed and estimated through oil red O staining. There were no surviving animals after the experiment.

## Transcriptome sequencing and data preparation

Venous blood from pigs was placed in a tube containing EDTA, red blood cell lysate was added, and the sediment was taken after repeated blowing and centrifugation. We extracted RNA as described in a previous report (*Hua et al., 2018*) and assessed its integrity using the Bioanalyzer 2100 system (Agilent Technologies, Santa Clara, CA, USA) with the RNA Nano 6000 Assay Kit as previously described (*Nawaz et al., 2018*). As input material for RNA sample preparation, 1 μg of RNA was used per sample. According to the manufacturer's instructions, sequencing libraries were generated using the NEBNext® UltraTM RNA Library Prep Kit for Illumina® (NEB, Ipswich, MA, USA). For each sample, index codes were added to the attribute sequences. PCR products were purified using an AMPure XP system, and the library quality was assessed using the Agilent Bioanalyzer 2100 system. Next, the index-coded samples were clustered using the TruSeq PE Cluster Kit v3-cBot-HS (Illumina, San Diego, CA, USA) on a cBot Cluster Generation System. A 150-bp pair-end read was generated from the libraries prepared on an Illumina NovaSeq platform after cluster generation. The biological replicate numbers of the control, HC+STZ, STZ+HC, HL+STZ, and STZ+HL groups were 4, 4, 3, 4, and 5, respectively. The technical replicate number of all samples was 2.

The statistical power of this experimental design, calculated in RNASeqPower, is 0.87. We processed in-house Perl scripts to process quality control raw data in fastq format. Clean data were obtained by removing adapters, poly-N sequences, and low-quality reads from raw data, which were the basis of all downstream analyses. Additionally, the Q20, Q30, and GC contents of the clean data were calculated.

## Reads mapping to the reference genome and quantification of gene expression levels

From the ensembl genomes website, we downloaded the reference genome (https://ftp.ensembl.org/pub/release-98/fasta/sus_scrofa_largewhite/dna/) and annotation files (https://ftp.ensembl.org/pub/release-98/gtf/sus_scrofa_largewhite/) for the *Sus scrofa*. HISAT2 v2.0.5 was used to build the index of the reference genome, and paired-end clean reads were aligned with it. The read numbers mapped to each gene were counted using

FeatureCounts v1.5.0-p3. Based on the gene length and read counts mapped to this gene, the expected number of fragments per kilobase of transcript sequence per million base pairs sequenced (FPKM) of each gene was calculated.

## Identification of differentially expressed genes in DM and AS

The gene expression dataset was normalized by $\log_2$ transformation with Benjamini–Hochberg correction to control the false discovery rate. A $p$ value < 0.05 and absolute $\log_2$ fold change (FC) ≥ 1 were regarded as threshold criteria for significant differentially expressed genes (DEGs) of interest. The DESeq2 R package (1.16.1) was used in this study to analyse differential expression between any two groups based on a negative binomial model. Benjamini and Hochberg's approach to controlling false discovery rates was used to adjust the resulting $p$ values. In DESeq2, DEGs with an adjusted $p$ value of 0.05 were considered.

## Gene ontology and KEGG pathway enrichment analyses

In the clusterProfiler R package, the gene length bias was corrected, and a corrected $p$ value of 0.05 was considered significant enrichment for DEGs. The statistical enrichment of DEGs was also tested among the KEGG (http://www.genome.jp/kegg/) pathways. An adjusted $p$ value < 0.05 was considered significant for all enrichment analyses.

## Protein–protein interaction analysis

The STRING database (*Szklarczyk et al., 2021*) was used to construct protein–protein interaction (PPI) networks of the proteins encoded by the identified DEGs. Network analysis and hub protein identification were performed using the cytoHubba modules in Cytoscape 3.7.0 (*Shannon et al., 2003*).

## Statistical analysis

The statistical software, statistical tests and significance thresholds applied for DEGs, Gene Ontology (GO) and KEGG analysis have been described above separately. Correlation analysis of blood glucose levels and gene expression was performed. Comparisons of multiple groups of measurement data were performed with one-way ANOVA, and the subsequent pairwise comparison adopted the Bonferroni method. A $p$ value < 0.05 was considered significant for the analysis.

# RESULTS

## Identification of DEGs common to HC+STZ *vs* control, STZ+HC *vs* control, HL+STZ *vs* control, and STZ+HL *vs* control

The blood glucose levels of pigs after modelling were 3.20 ± 0.91, 23.23 ± 3.28, 2.93 ± 0.92, 24.50 ± 2.68, and 3.80 ± 0.71 mmol/L in the control, HC+STZ, STZ+HC, HL+STZ, and STZ+HL groups, respectively. Abdominal aorta atherosclerotic plaques were found in all pigs from the HC+STZ group. In other models, only three cases from the STZ+HL group showed plaques (*n* = 1) or lipid streaks (*n* = 2). The significant DEGs from Ba-Ma mini pigs were identified using applied combinatorial statistical methods. A total of 103 DEGs were identified as common between the HC+STZ and STZ+HC groups, which were

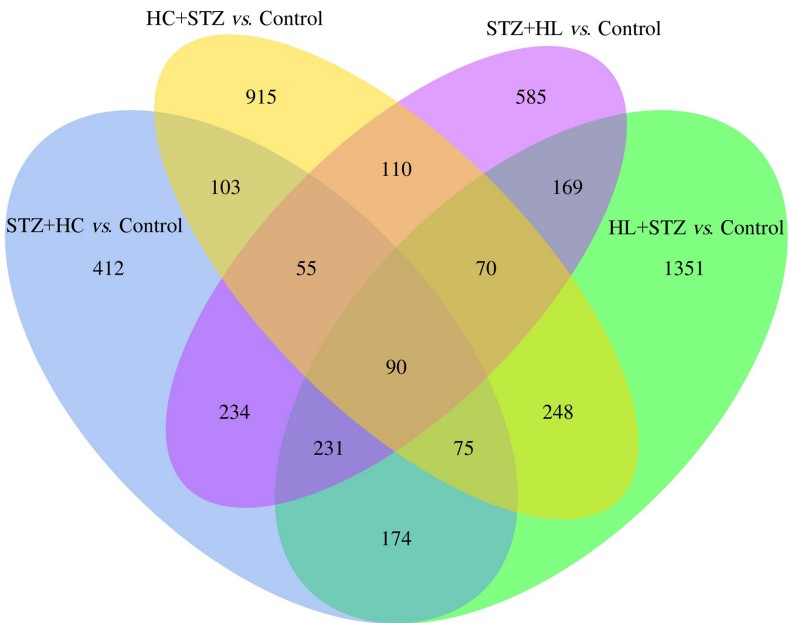

**Figure 1 The common DEGs between the HC+STZ, STZ+HC, HL+STZ, and STZ+HL groups.** A total of 103 transcripts were identified as common between the DEGs of the HC+STZ and STZ+HC groups. A total of 915 and 412 DEGs were specifically expressed in the HC+STZ and STZ+HC groups, respectively. HC+STZ: pigs were fed a high-cholesterol/lipid diet for 9 months and injected with STZ at 3 months (*n* = 4). STZ+HC: pigs were injected with STZ first and then fed a high-cholesterol/lipid diet for 9 months (*n* = 3). HL+STZ: pigs were fed a high-lipid diet for 9 months and injected with STZ at 3 months (*n* = 4). STZ+HL: pigs were injected with STZ first and then fed a high-lipid diet for 9 months (*n* = 5). The biological replicate numbers of the control, HC+STZ, STZ+HC, HL+STZ, and STZ+HL groups were 4, 4, 3, 4, and 5, respectively. The technical replicate number of all samples was 2.

different from the HL+STZ and STZ+HL groups (Fig. 1). The numbers of consistently upregulated and downregulated genes in both groups were 44 and 42, respectively. Moreover, 17 genes were upregulated in the HC+STZ group and downregulated in the STZ+HC group. As shown in Fig. 2, the distribution of 103 DEGs was different not only between the groups but also within the groups, suggesting the potential application value of DEGs in distinguishing and refining the molecular mechanism of DM and AS. According to the hyperglycaemia and plaque results, it is speculated that these 103 DEGs may be involved in the common mechanism of DM and AS, and these genes were included in the further follow-up analysis.

## Significant GO terms and pathways

The enrichment analysis of identified genes was used to assess the biological processes (BPs), cellular components (CCs), and molecular functions (MFs) in which the enriched genes participated. The top three enriched BPs of DEGs in HC+STZ *vs* control or STZ+HC *vs* control were regulation of catabolic process, cellular catabolic process, and organic substance catabolic process ($p < 0.05$, Fig. 3A) or antigen processing/presentation, carbohydrate derivative metabolic process, and immune response ($p < 0.05$, Fig. 4A),

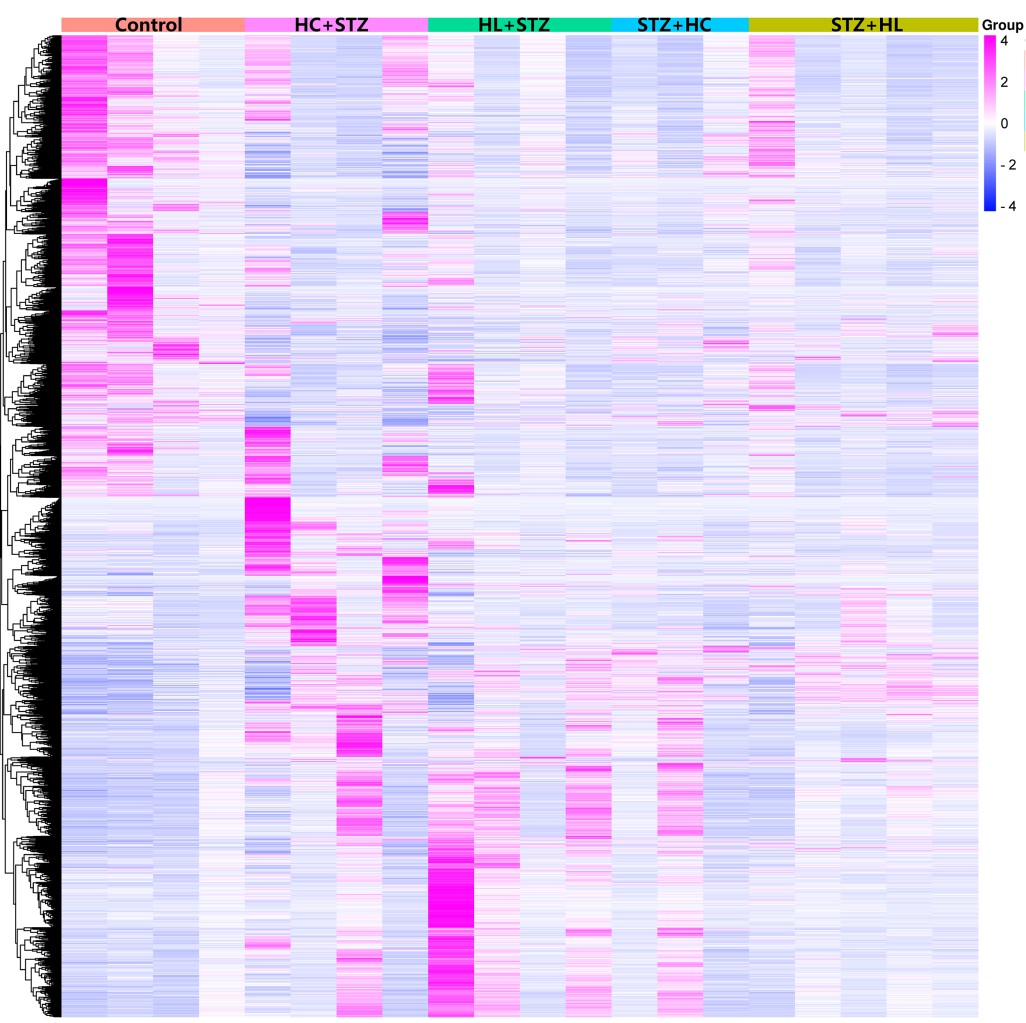

**Figure 2 Heatmap of the DEGs in the different groups.** Annotations on the top of the heatmap show clustering of the samples. The Z score represents the normalized value of the differential gene FPKM. The closer the color is to rose red, the higher the gene expression is. In contrast, the closer the color is to blue, the lower the gene expression is. HC+STZ: pigs were fed a high-cholesterol/lipid diet for 9 months and injected with STZ at 3 months ($n = 4$). STZ+HC: pigs were injected with STZ first and then fed a high-cholesterol/lipid diet for 9 months ($n = 3$). HL+STZ: pigs were fed a high-lipid diet for 9 months and injected with STZ at 3 months ($n = 4$). STZ+HL: pigs were injected with STZ first and then fed a high-lipid diet for 9 months ($n = 5$).

respectively. The top three significant CCs of DEGs in HC+STZ *vs* control or STZ+HC *vs* control were proteasome core complex, proteasome complex, and endopeptidase complex ($p < 0.05$, Fig. 3B) or microtubule organizing centre, preribosome, and chromosome ($p < 0.05$, Fig. 4B), respectively. Carboxy-lyase activity, purine nucleoside binding, and GTP binding ($p < 0.05$, Fig. 3C) or phosphoric diester hydrolase activity, sulfuric ester hydrolase activity, and hydrolase activity ($p < 0.05$, Fig. 4C) were identified as significant MFs. The most significantly enriched pathways analysed through KEGG were endocytosis, regulation of actin cytoskeleton, fructose, mannose metabolism ($p < 0.05$, Fig. 3D) and DNA replication and the toll-like receptor signalling pathway ($p < 0.05$, Fig. 4D).

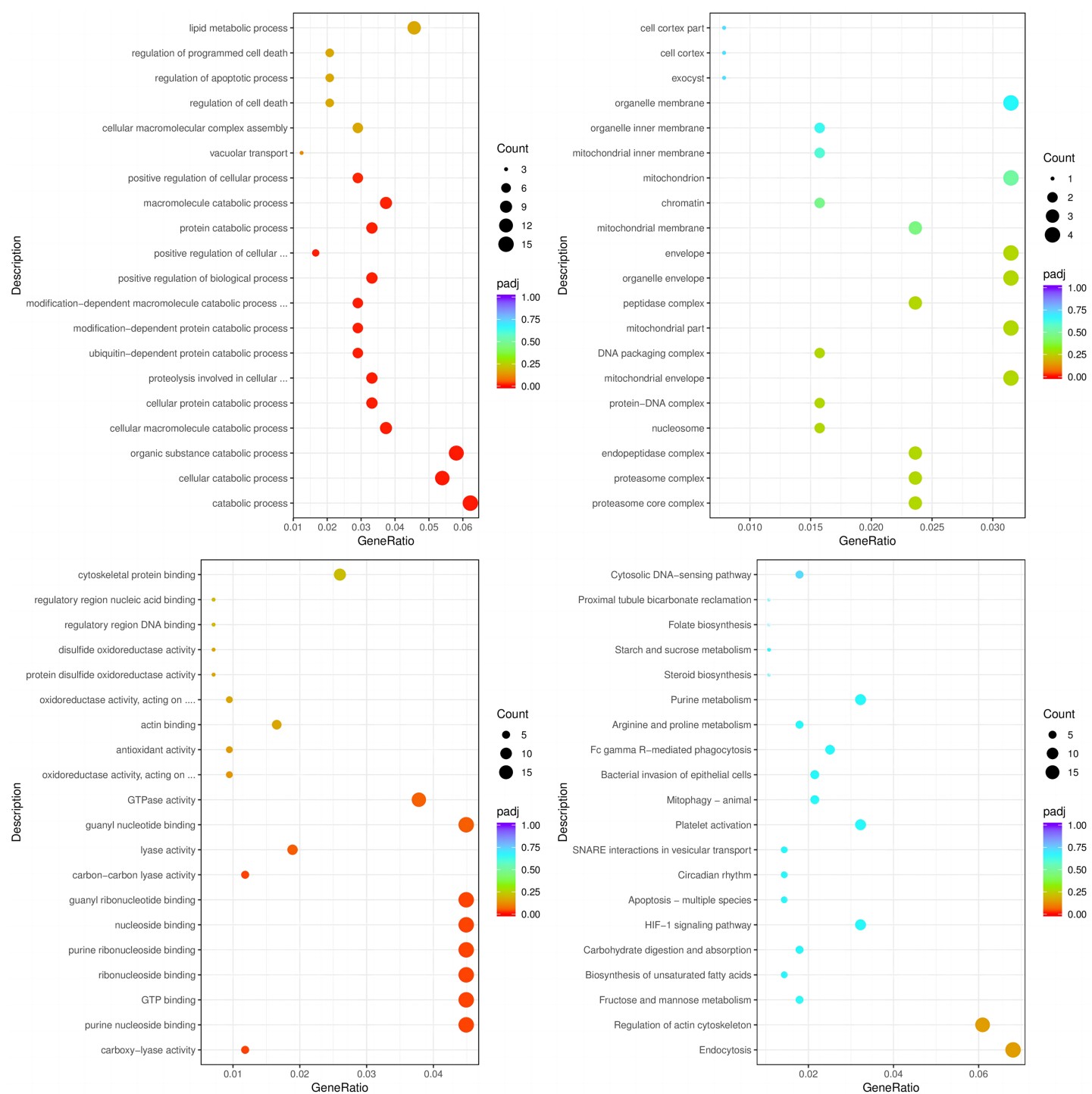

**Figure 3  The BP, CC, and MF terms or pathways enriched in the 915 genes from the HC+STZ group were analysed by GO and KEGG analyses.** HC+STZ: pigs were fed a high-cholesterol/lipid diet for 9 months and injected with STZ at 3 months (*n* = 4). GeneRatio, the ratio of the number of genes annotated to this entry in the selected gene set to the total number of genes annotated to this entry in this species. Padj, also known as the *p* value correction value.

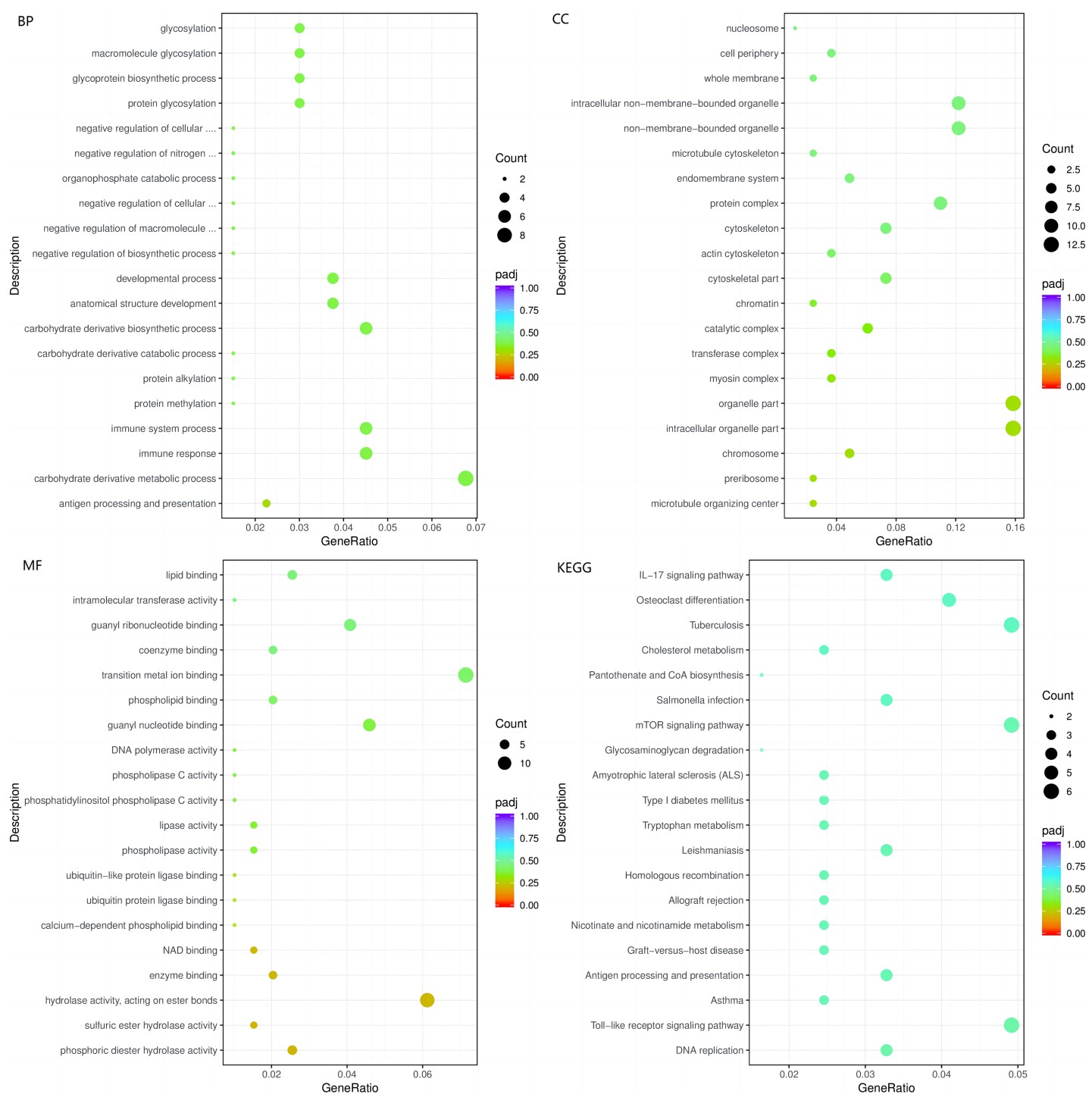

**Figure 4 The BP, CC, and MF terms or pathways enriched in the 412 genes from the STZ+HC group were analysed by GO and KEGG analyses.** STZ+HC: pigs were injected with STZ first and then fed a high-cholesterol/lipid diet for 9 months (*n* = 3). GeneRatio, the ratio of the number of genes annotated to this entry in the selected gene set to the total number of genes annotated to this entry in this species. Padj, also known as the *p* value correction value.

Collectively, these data suggest that the DEGs in the HC+STZ *vs* control and STZ+HC *vs* control groups were mainly enriched in catabolic pathways, metabolic processes, immune responses and DNA replication.

The top three enriched BPs were regulation of response to biotic stimulus, immune response, and immune system process ($p < 0.05$, Fig. 5A). The top three significant CCs of the 103 DEGs common to HC+STZ and STZ+HC were extracellular region, proteinaceous extracellular matrix, and viral capsid ($p < 0.05$, Fig. 5B). Chemokine activity, chemokine receptor binding, G-protein coupled receptor binding and others were identified as significant MFs ($p < 0.05$, Fig. 5C). The most significantly enriched pathways were influenza A, hepatitis C, and measles ($p < 0.05$, Fig. 5D). These results demonstrated that 103 common DEGs were mainly enriched in immune response and regulation.

### Identification of hub proteins

The PPI network was first constructed by retrieving the global and internal interactions of the 103 common DEGs between the HC+STZ and STZ+HC groups from the STRING database (Fig. 6A). The PPI network consisted of 648 nodes (72 nodes from the common DEGs) and 2,106 edges, revealing the top 10 hub proteins, namely, ISG15, IRG6, IRF7, IFIT3, MX1, UBE2L6, DDX58, IFIT2, USP18, and IFI44L (Fig. 6B). The interactions within 103 common DEGs (14 nodes and 31 edges) were further displayed using the PPI network shown in Fig. 6C. Unexpectedly, MX1 and UBE2L6 were the intersection of internal and global PPI networks of 103 common DEGs, suggesting their key role in the comorbidity mechanism of DM and AS (Figs. 6A and 6C). Our results revealed that MX1 and UBE2L6 have different roles, in addition to the traditional anti-virus activity and ubiquitination degradation of proteins; furthermore, the data indicate that the above biological processes may be also involved in the pathophysiology of DM and AS.

### Differential expression of MX1 and UBE2L6 and their contributions to DM and AS

The expression of MX1 and UBE2L6 was 507.22 ± 342.56 and 96.99 ± 49.92 in the HC+STZ group, respectively, which was significantly higher than that in the control group ($p = 0.0178$ and $p = 0.0188$, respectively; Figs. 7A and 7B). In contrast, the expression of MX1 and UBE2L6 in the other group with stable hyperglycaemia was only 31.10 ± 19.21 and 28.21 ± 14.30, similar to that in the control group ($p < 0.05$ and $p < 0.05$, respectively; Figs. 7A and 7B). The Sankey diagram traced the glucose levels and gene expression of models with or without AS. Obviously, in the HC+STZ group, there were two cases with atherosclerotic plaques exceeding 50% of the lumen and two other cases with plaques that were less than half of the lumen, with serious hyperglycaemia. The coexistence of DM and AS was related to the overexpression of MX1 and UBE2L6 in the HC+STZ group (Figs. 7C and 7D).

### MX1- and UBE2L6-related PPI networks for calcium and micronutrients

A total of 13,802 and 382 potential candidate genes were identified in the GeneCards databases. Calcium- and micronutrient-related PPI networks were constructed by

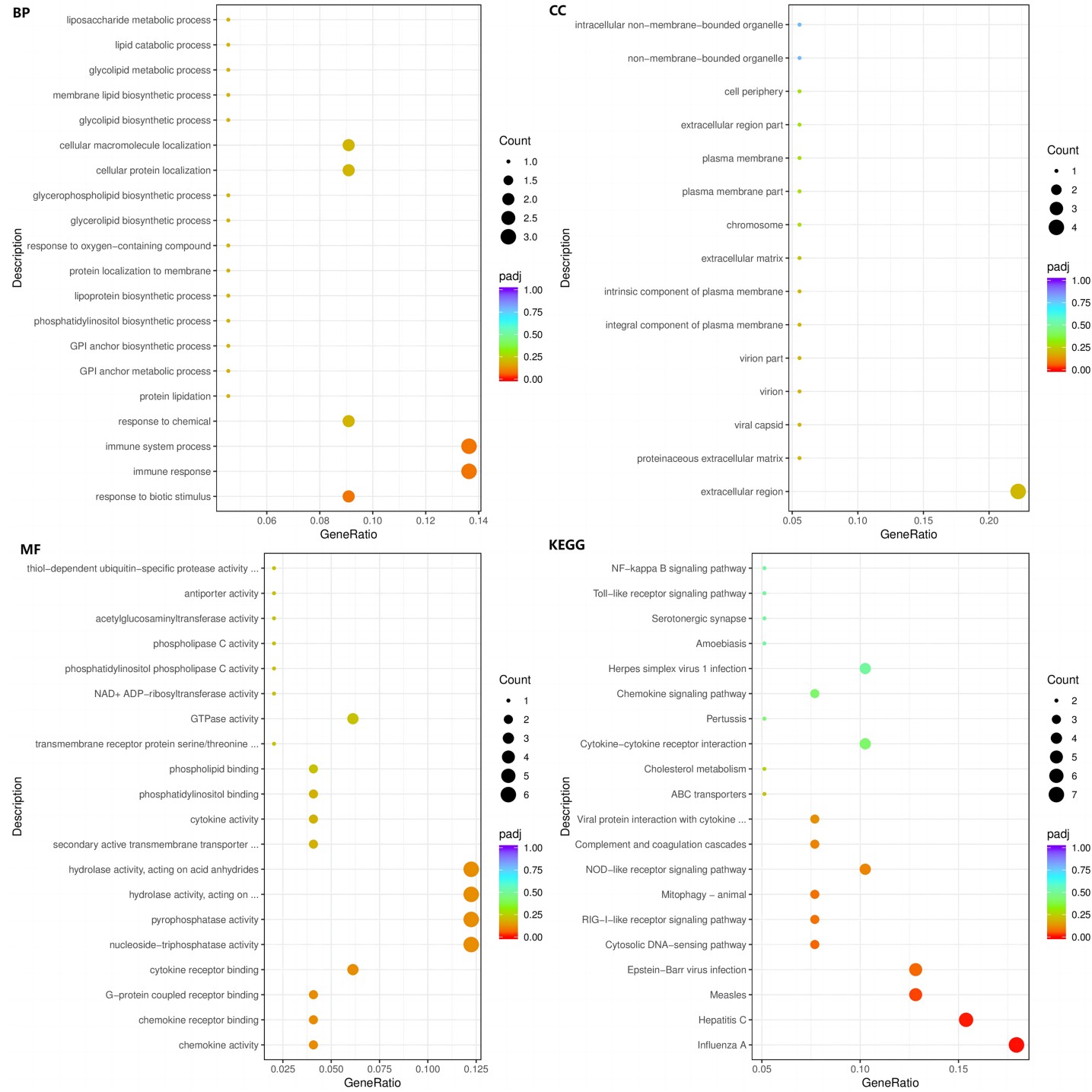

**Figure 5** **The BP, CC, and MF terms or pathways enriched in the 103 genes common to the HC+STZ and STZ+HC groups were analysed by GO and KEGG analyses.** HC+STZ: pigs were fed a high-cholesterol/lipid diet for 9 months and injected with STZ at 3 months ($n = 4$). STZ+HC: pigs were injected with STZ first and then fed a high-cholesterol/lipid diet for 9 months ($n = 3$). GeneRatio, the ratio of the number of genes annotated to this entry in the selected gene set to the total number of genes annotated to this entry in this species. Padj, also known as the *p* value correction value.

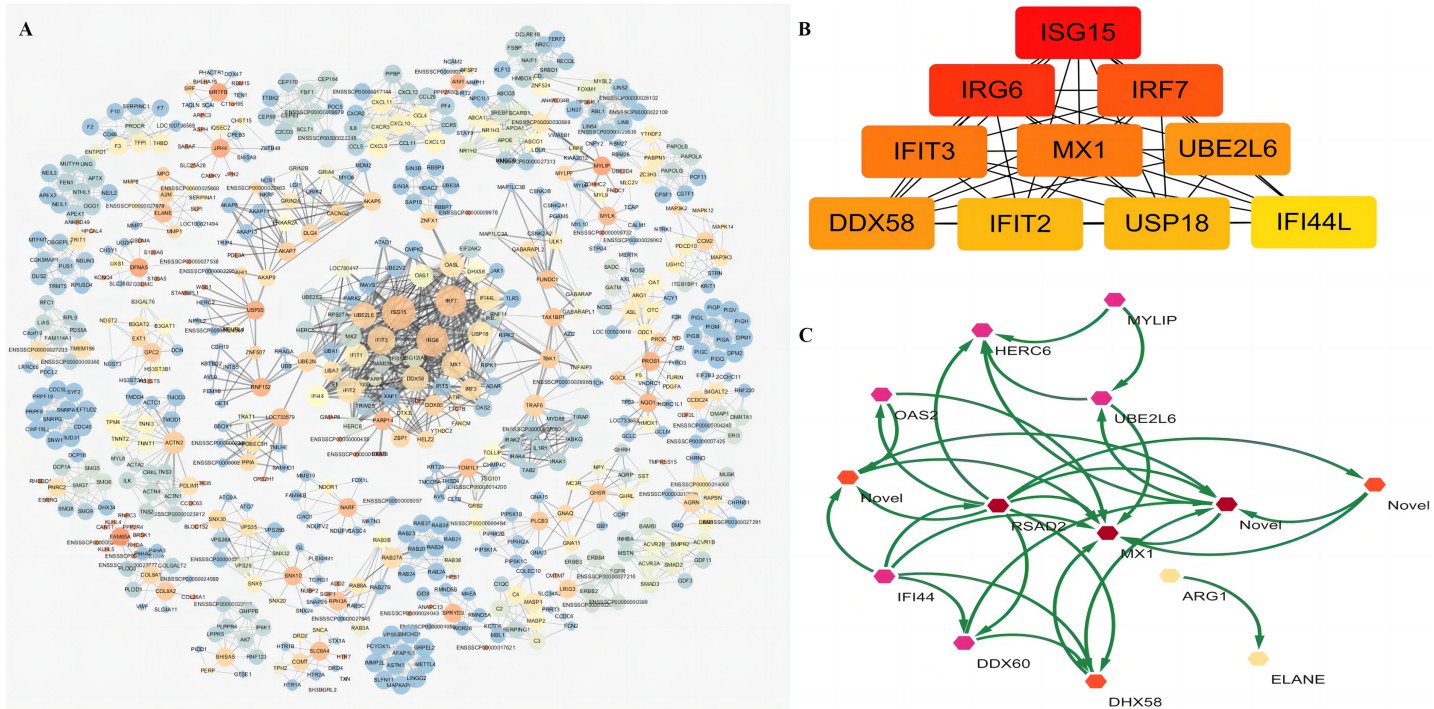

**Figure 6 PPI analysis of 103 common DEGs of HC+STZ *vs* control and STZ+HC *vs* control.** (A) The global PPI network, which consists of 648 nodes and 2,106 edges, was constructed by retrieving the global and internal interactions of the 103 common DEGs from the STRING database. (B) The top 10 hub proteins in the global PPI network were screened using the CytoHubba plugin of the Cytoscape software using the Degree method. The darker the colour is in the color gradient, the higher the coreness is. (C) Internal PPI network of 103 common DEGs, consisting of 14 nodes and 31 edges. HC+STZ: pigs were fed a high-cholesterol/lipid diet for 9 months and injected with STZ at 3 months ($n$ = 4). STZ+HC: pigs were injected with STZ first and then fed a high-cholesterol/lipid diet for 9 months ($n$ = 3).

retrieving the interactions of the top 1,998 candidates (if applicable) with MX1 and UBE2L6 from the STRING database. The calcium-related PPI network consisted of 1,744 nodes and 33,764 edges. The micronutrient-related PPI network consisted of 342 nodes and 2,825 edges. The calcium-related PPI network in which MX1 and UBE2L6 participate contained 58 nodes and 424 edges (Fig. 8A). The micronutrient-related PPI network in which MX1 and UBE2L6 participate contained 18 nodes and 58 edges (Fig. 8B). The 13 common proteins were MX1, UBE2L6, IFNG, CD4, TNF, VEGFA, CXCL10, PRDX5, F2, IL10, CCL2, IL6, and TP53. The most significantly enriched KEGG pathways were glutathione metabolism, pyrimidine metabolism, purine metabolism, and metabolic pathways. The data revealed that MX1- and UBE2L6 have a close relation to calcium and micronutrient metabolic pathways.

# DISCUSSION

The higher prevalence and gravity establish a superficial connection between DM and AS. On the one hand, several studies have been devoted to the genetic aetiology of DM (*Buraczynska & Zakrocka, 2021*; *Deaton et al., 2021*; *Dieter et al., 2021*) and AS (*Posadas-Sanchez et al., 2017*) and aimed to link preselected individual genes with AS in DM patients. The results showed associations between a matrix metalloproteinase-3 gene polymorphism (rs3025058) (*Pleskovic et al., 2017*), phosphoprotein 1 (rs4754) (*Pleskovic*

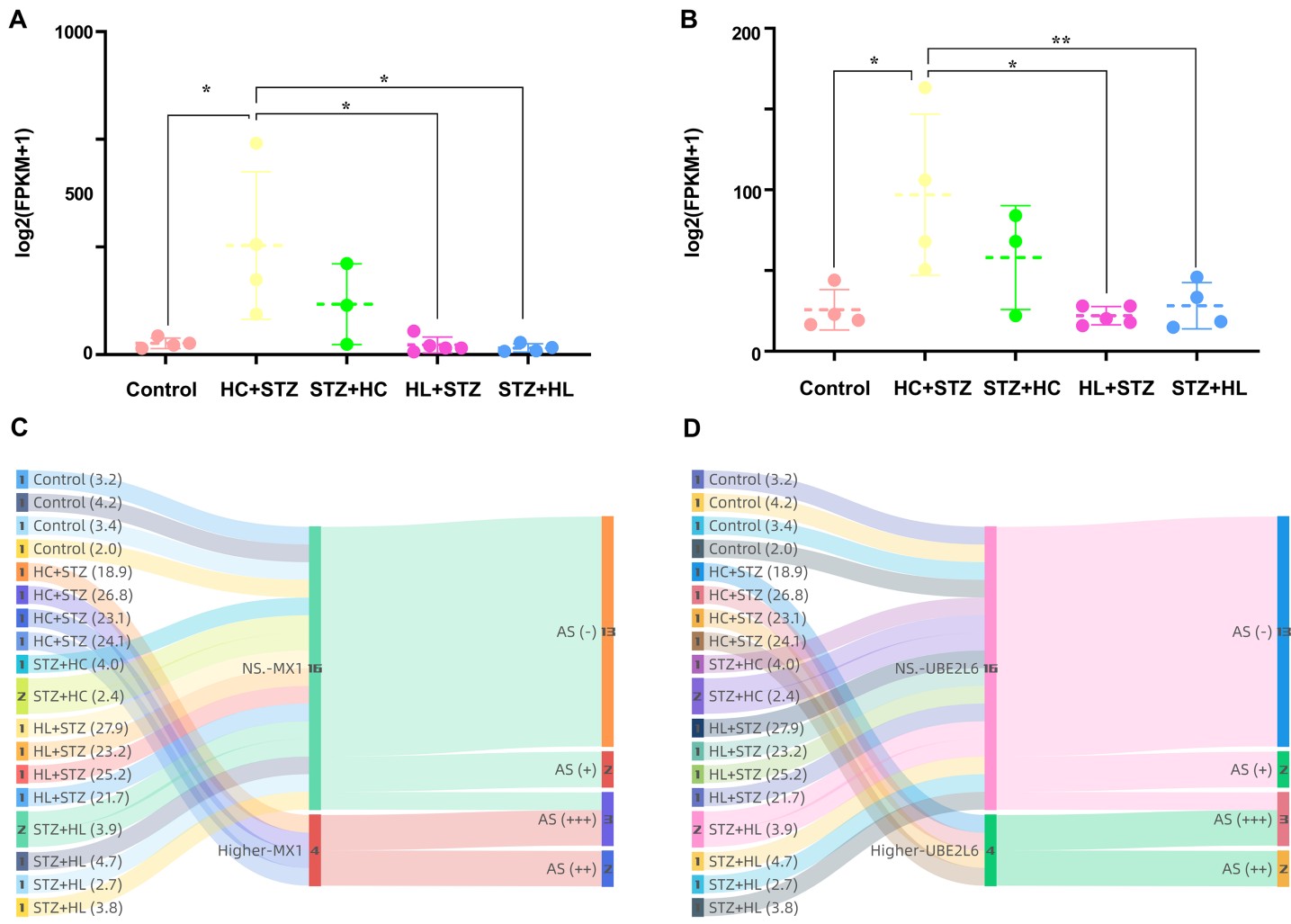

**Figure 7** **The expression of MX1 and UBE2L6 links DM and AS.** (A and B) The expression of MX1 and UBE2L6 was significantly increased in the HC+STZ group compared with the other groups ($*p < 0.05$; $**p < 0.01$). Data shown as scatter plots with mean ± standard deviation. (C and D) The association between the presence or severity of AS of the abdominal aorta and overexpression of MX1 or UBE2L6 related to higher levels of glucose was traced by a Sankey diagram. The numbers in each colour bar represent the number of samples for the corresponding classification. The numbers in parentheses after each group label are their fasting blood glucose values (measured in mmol/l, expressed as the mean). HC+STZ: pigs were fed a high-cholesterol/lipid diet for 9 months and injected with STZ at 3 months ($n = 4$). STZ+HC: pigs were injected with STZ first and then fed a high-cholesterol/lipid diet for 9 months ($n = 3$). HL+STZ: pigs were fed a high-lipid diet for 9 months and injected with STZ at 3 months ($n = 4$). STZ+HL: pigs were injected with STZ first and then fed a high-lipid diet for 9 months ($n = 5$). At higher MXI/UBE2L6 levels, the expression levels of the corresponding genes were significantly higher than those in the control group. NS. MXI/UBE2L6, the expression levels of the corresponding genes were similar to those in the control group. AS (−), no fatty streak or atherosclerotic plaque; AS (+), fatty streak; AS (++), atherosclerotic plaque is less than half of the lumen; AS (+++), atherosclerotic plaque is more than half of the lumen.

*et al., 2018*), a SOX6 gene polymorphism (rs16933090) (*Pleskovic et al., 2016*), the renin-angiotensin-aldosterone system, apolipoprotein E, methylenetetrahydrofolate reductase and proinflammatory genes (*Ramus & Petrovic, 2019*). Unfortunately, an aerial view of genes involved in DM and AS separately or crosswise is lacking, and the causal link, detailed mechanism and signalling pathways are still unknown. On the other hand, although chronic inflammation is thought to be the main pathological driver of a variety of metabolic diseases, it is unclear whether DM and AS maintain common inflammatory

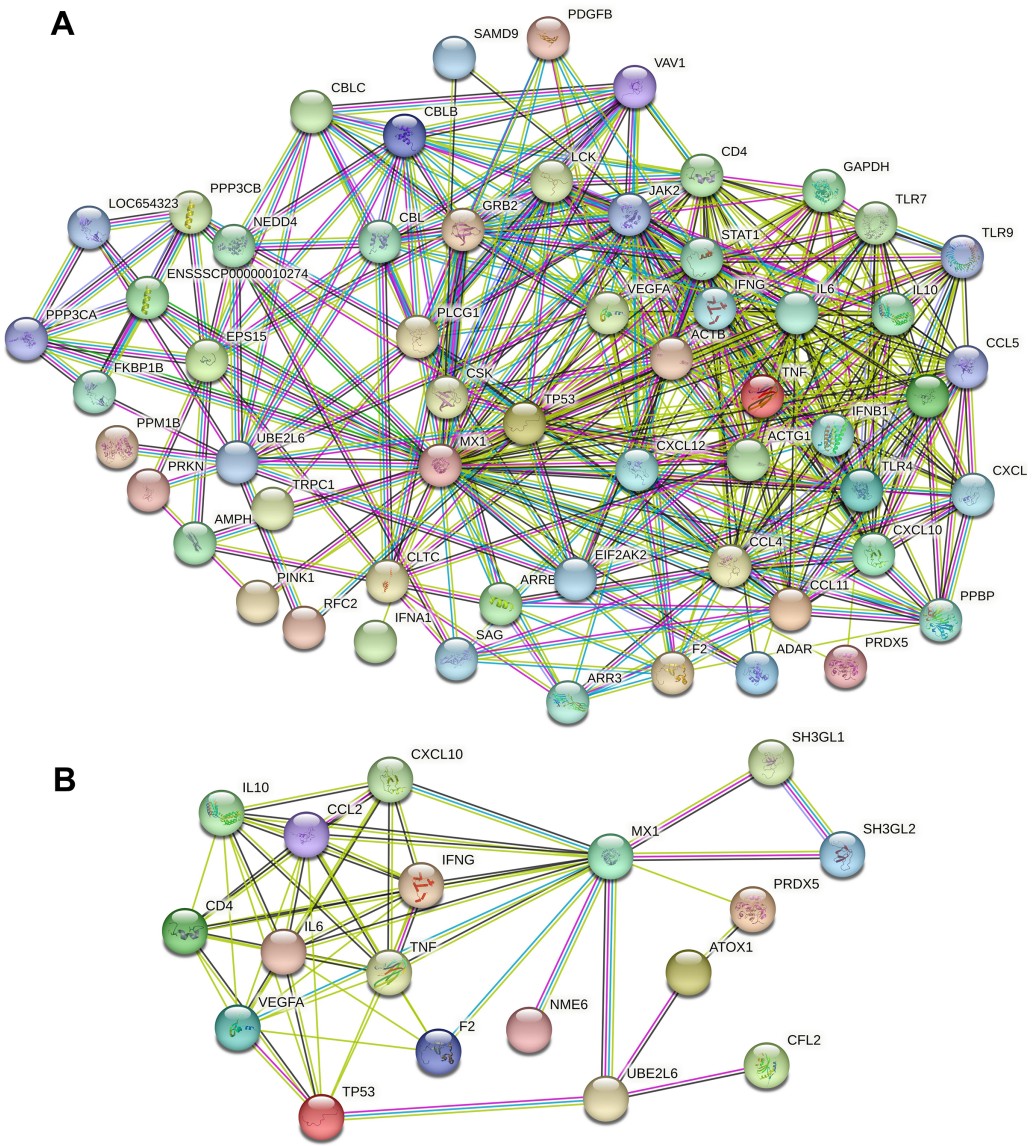

**Figure 8 PPI analysis of calcium and micronutrient potential candidate genes (*n* = 13,802 and 382) with MX1 and UBE2L6.** (A) The calcium-related PPI network in which MX1 and UBE2L6 participate contained 58 nodes and 424 edges. (B) The micronutrient-related PPI network in which MX1 and UBE2L6 participate contained 18 nodes and 58 edges.

mechanisms. MAMPs were newly classified as harmful metabolic-stimulated inflammatory responses that cause sequential tissue damage by stimulating pattern recognition receptors (*Wang et al., 2020*). Altogether, MAMPs can incite proinflammatory signalling pathways, drive the production of proinflammatory mediators and initiate or participate in the process of metabolic diseases, including DM and AS (*Wang et al., 2020*). However, the detailed MAMPs that characterize or cross characterize DM and AS are obscure. Interestingly, we first found key molecules involved in metabolism and inflammation that cause the comorbidity of DM and AS.

In the present research, we identified the common DEGs in the different groups of model pigs, especially in the HC+STZ and STZ+HC groups, in which AS was more frequent and serious. There were 103 DEGs common to the HC+STZ vs control and STZ +HC vs control groups. GO and pathway analysis revealed the crucial roles of related modules and signalling pathways, such as virus infection, enzyme activity, toll-like pathways and mitophagy. The PPI network was constructed by retrieving the interactions of the common DEGs from the STRING database, which revealed the top 10 hub proteins predicted to promote DM and AS. Among them, MX1 and UBE2L6 were also found in the internal PPI network of the 103 common DEGs, which may be candidate biomarkers and potential therapeutic targets of DM and AS individually or in combination.

In the experiment involving pig models with a high-cholesterol, high-fat diet and STZ injection, MX1 and UBE2L6 were successfully screened out as much as possible to reduce the confounding of other factors and diseases. Moreover, we reanalyzed the gene expression of MX1 and UBE2L6 in all groups. MX1 and UBE2L6 expression in the HC +STZ group was significantly higher than that in the other groups. The axis of hyperglycaemia-higher MX1 and UBE2L6 expression-serious AS was present, which also highlighted the importance of MX1 and UBE2L6 in the comorbidity of DM and AS. Of those factors identified, UBE2L6 usually acts as a ubiquitin-conjugating enzyme and was reported to be related to lipid metabolism (*Wei et al., 2021*), apoptosis of *Mycobacterium tuberculosis*-infected macrophages (*Gao et al., 2021*), and autophagy attenuation in oesophageal cancer cells (*Falvey et al., 2017*). MX1 was reported to be changed as an antiviral gene and immune response factor in patients with hypogammaglobulinemia (*Wirz et al., 2020*), lupus nephritis (*Cao et al., 2020*), Zika virus infection (*Anderson et al., 2021*), lupus erythematosus (*Xiang et al., 2021*) and so on.

Interestingly, calcium influx may bridge the coexistence of DM and AS with MAMPs. Accumulating evidence has established a connection between obesity-induced complications and transient receptor potential vanilloid type 1 (TRPV1), which serves as a ligand-gated nonselective cation channel. This notion was partly supported by the significant decrease in the mean concentrations of calcium in the patients' serum samples (*Rao et al., 2022*). Theoretically, as a GTP-binding protein, MX1 can also mediate calcium influx through G protein pathways, although this underlying mechanism and its effect on DM and AS have not been explored. Calcium- and micronutrient-related PPI networks were constructed by retrieving the interactions of the candidates with MX1 and UBE2L6, from which 13 common proteins were screened. The most significantly enriched pathways were glutathione metabolism, pyrimidine metabolism, purine metabolism, and metabolic pathways. Numerous studies have shown that glutathione metabolism participates in the pathological mechanism of glucose and lipid metabolism-related diseases by regulating redox balance (*Huang et al., 2018*) and transcription of pancreatic and duodenal homeobox 1 (Pdx1) (*Yun et al., 2019*).

These two compounds identified from the interaction networks may contribute to the study of common mechanisms in DM and AS by virtue of their internal biological roles associated with MAMPs and calcium and micronutrient metabolism, although further validation is also needed. Of course, the specific mechanism of inflammation and

micronutrient metabolism involving MX1 and UBE2L6 in DM and AS still needs further research, and the roles of MAMPs and their pathways still need to be confirmed by population omics research.

## CONCLUSIONS

Atherosclerotic lesions were observed in the HC+STZ and STZ+HC groups, although the serum glucose concentrations only increased significantly in the HC+STZ and HL+STZ groups. A total of 103 DEGs were identified as common between the HC+STZ and STZ +HC groups, which were different from other groups and mainly enriched in immune response and regulation. The global and internal PPI network consisted of 648 and 14 nodes, respectively. The intersection of the nodes from 10 hub proteins of the global network and internal PPI network was MX1 and UBE2L6, which suggests their key role in the comorbidity mechanism of DM and AS. This inference was partly verified by the overexpression of MX1 and UBE2L6 in the HC+STZ group but not in the other groups. Further calcium- and micronutrient-related enriched KEGG pathway analysis supported that MX1 and UBE2L6 may affect the inflammatory response through micronutrient metabolic pathways, conceptually named metaflammation. Collectively, MX1 and UBE2L6 may be potential biomarkers for DM and AS that may reveal metaflammation-related aspects of the pathological process, although more detailed studies are needed.

### Funding

This work was supported by the National Key R&D Program of China (No. 2022YFB4702602), the Beijing Natural Science Foundation (7222319), the Beijing Municipal Science & Technology Commission (Z21100002921047), and the National Natural Science Foundation of China (NSFC 82001808). The funders had no role in study design, data collection and analysis, decision to publish, or preparation of the manuscript.

### Grant Disclosures

The following grant information was disclosed by the authors:
National Key R&D Program of China: 2022YFB4702602.
Beijing Natural Science Foundation: 7222319.
Beijing Municipal Science & Technology Commission: Z21100002921047.
National Natural Science Foundation of China: NSFC 82001808.

### Competing Interests

The authors declare that they have no competing interests.

### Author Contributions

- Guisheng Wang conceived and designed the experiments, prepared figures and/or tables, authored or reviewed drafts of the article, and approved the final draft.

- Rongrong Hua conceived and designed the experiments, analyzed the data, prepared figures and/or tables, authored or reviewed drafts of the article, and approved the final draft.
- Xiaoxia Chen conceived and designed the experiments, prepared figures and/or tables, authored or reviewed drafts of the article, and approved the final draft.
- Xucheng He performed the experiments, prepared figures and/or tables, and approved the final draft.
- Yao Dingming performed the experiments, prepared figures and/or tables, and approved the final draft.
- Hua Chen performed the experiments, prepared figures and/or tables, and approved the final draft.
- Buhuan Zhang performed the experiments, prepared figures and/or tables, and approved the final draft.
- Yuru Dong performed the experiments, prepared figures and/or tables, and approved the final draft.
- Muqing Liu performed the experiments, analyzed the data, prepared figures and/or tables, and approved the final draft.
- Jiaxiong Liu performed the experiments, prepared figures and/or tables, and approved the final draft.
- Ting Liu performed the experiments, prepared figures and/or tables, and approved the final draft.
- Jingwei Zhao performed the experiments, prepared figures and/or tables, and approved the final draft.
- Yu Qiong Zhao performed the experiments, prepared figures and/or tables, and approved the final draft.
- Li Qiao conceived and designed the experiments, prepared figures and/or tables, authored or reviewed drafts of the article, and approved the final draft.

## Animal Ethics

The following information was supplied relating to ethical approvals (*i.e.*, approving body and any reference numbers):

Ethics Committee of Third Medical Center of Chinese PLA General Hospital.

## Data Availability

The data is available at GEO: GSE193781.

## Supplemental Information

Supplemental information for this article can be found online at http://dx.doi.org/10.7717/peerj.16975#supplemental-information.

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
