# Peer review of "MX1 and UBE2L6 are potential metaflammation gene targets in both diabetes and atherosclerosis"

_PeerJ, doi:10.7717/peerj.16975_

## Round 0.1 · original submission · Major Revisions

Please address the concerns of all reviewers and amend the manuscript accordingly.

Reviewer 1 ·

Basic reporting

NA

Experimental design

Authors should mention the blood glucose level of pigs after developing diabetes by STZ.

Validity of the findings

NA

Additional comments

NA

·

Basic reporting

The manuscript presents original research findings related to gene expression profiles in a specific animal model, which can contribute to the understanding of diabetes mellitus (DM) and atherosclerosis (AS).

The abstract does not effectively summarize the key findings and significance of the study. Similarly, the conclusion lacks conciseness and does not effectively summarize the main results and their implications.

The manuscript contains numerous grammatical errors and awkward phrasing, which impacts overall readability and comprehension.

The manuscript contains sentences and phrases that are repetitive and redundant, which can be condensed for clarity.

The manuscript lacks a consistent citation style, and references should follow a specific format.

Experimental design

The study employs various data analysis methods, including gene ontology (GO), KEGG pathway analysis, and protein-protein interaction (PPI) network analysis, demonstrating a comprehensive approach to understanding the molecular mechanisms. Identifying hub proteins (MX1 and UBE2L6) and their potential roles in the comorbidity of DM and AS is noteworthy and could have implications for further research and potential therapeutic targets. The study attempts to bridge the fields of metabolism, inflammation, and genetics, offering a potentially valuable interdisciplinary perspective. Further, it explores common inflammation/metabolism biomarkers that may shed light on the development and progression of DM and AS, which is an important research goal.

The manuscript mentions the use of Ba-Ma mini pigs as an animal model. However, it would be beneficial to provide more information about why this specific animal model was chosen, its relevance to human DM and AS, and any potential limitations associated with this choice.

The text mentions the creation of different experimental groups, such as the HC+STZ, STZ+HC, HL+STZ, and STZ+HL groups. It would be helpful to explain the rationale behind these groupings and how they relate to the research questions. Additionally, details about the number of animals in each group and their characteristics (e.g., age, sex) should be included.

The manuscript briefly mentions statistical methods but does not provide details. It's important to describe the statistical tests used to analyze the data, including the significance thresholds applied. Additionally, mention if corrections for multiple comparisons were made.

Validity of the findings

Please validate if gene expression data are central to the findings, and describe the validation methods used to confirm the gene expression changes. The author can involve techniques like qPCR or immunohistochemistry.
Consider whether future longitudinal studies might be needed to confirm the observed associations and assess causality over time.

Please explain and discuss the biological relevance of the pathways identified. Are the identified pathways consistent with existing knowledge of diabetes and atherosclerosis?

Reviewer 3 ·

Basic reporting

Please see Additional Comments.

Experimental design

Please see Additional Comments.

Validity of the findings

Please see Additional Comments.

Additional comments

The authors analyzed gene expression profiles of Ba-Ma mini pig models with diabetes and atherosclerosis, identifying some genes that could be useful biomarkers or targets for these diseases. The current study would be much improved if the authors address the following concerns:


------[Major Concerns about FIGURES, METHODS, RESULTS, and/or CONCLUSIONS]
1. In all FIGURES, it would be clear and more readable to BOTH provide figures with high resolution AND expand on figure legends by explaining the meanings of colors, groups, lines, and abbreviations. These revisions would greatly help readers to understand the results and their implications easily and efficiently. For example,
1.1 In all FIGURES' bar graphs, it would be more informative to display individual data points; in other words, please replace bar graphs by EITHER scatter plots with bars OR scatter plots (a pattern like PMID: 34537192, PMID: 37046252, and PMID: 37452367). Bar graphs have been shown to be misleading, because they cannot reveal variation/dispersion within data; instead, scatter plots with bars could be acceptable and scatter plots would be preferable (as confirmed by PMID: 25901488 and PMID: 28974579).
1.2 In all FIGURES' legends, it would be more rigorous to mention BOTH the sample size (the number of data points OR how many samples/patients were included) AND whether the data points were technical or biological replicates.
1.3 In all FIGURES' legends, it would be more rigorous to mention how the authors reported the data error (variation/dispersion): standard deviation (SD), confidence intervals (CI), or standard error of the mean (SEM, which would be not preferable).
1.4 In the legend of all FIGUREs, it would be clearer (easier to understand) to mention full names of abbreviations like "HC+STZ, STZ+HC, HL+STZ, and STZ+HL".
1.5 In the legend of Figure 2, it would be more informative to mention the meaning of the x-axis, y-axis, and red & green colors.
1.6 In the legend of Figures 3–5, it would be clearer (easier to understand) to briefly explain the meaning of "GeneRatio", "Count", and "padj" in Figures 3–5.
1.6 In the legend of Figure 6, it would be more informative to mention whose "DEGs" those genes were (from which groups?)
1.7 In the legend of Figure 7, the legends seem inconsistent with the panels. Furthermore, please expand on elements in the panels C & D.

2. In ABSTRACT:
2.1 In Background, it would be more informative to introduce "metaflammation", which is a keyword in TITLE.
2.2 In Methods, it would be more rigorous and clearer (easier to understand) to briefly explain how the authors established "DM and AS in pigs" (which was mentioned in Results but not explained in Methods). In other words, please point out which group was DM and which was AS in Methods.
2.3 In Methods ("... using different combinations of a high-cholesterol/lipid (HC) diet, a high-lipid (HL) diet, and streptozotocin (STZ) injection"), it would be more informative to list all "combinations" and mention the implication of each group (that is, which group was disease model and which one was control).
2.4 In Results ("A total of 103 differentially expressed genes (DEGs) were identified as common between DM and AS in pigs"), it would more informative to mention how many DEGs were upregulated and how many were downregulated.
2.5 In Results ("The most significantly enriched pathways were influenza A, hepatitis C, and measles"), it would be more rigorous to mention which genes (all DEGs, the upregulated, or the downregulated?) were used for the analysis. Likewise, this issue could happen to "The top 10 hub proteins, namely, ISG15, IRG6, IRF7, IFIT3, MX1, UBE2L6, DDX58, IFIT2, USP18, and IFI44L, drive aspects of DM and AS".
2.6 In Results ("The top 10 hub proteins ... MX1 and UBE2L6 were also included in the internal protein–protein interaction (PPI) network. The expression of MX1 and UBE2L6 was ..."), it would be more convincing to rewrite this sentence by explaining why the authors specially focused on "MX1 and UBE2L6", rather than some of the top 10 hub proteins. Also, please expand on the reason in the main text.

3. In RESULTS:
3.1 It would be clearer to end each paragraph in RESULTS with one sentence: "Together, these results suggest that ..." (a pattern like PMID: 37452367, PMID: 34715879, PMID: 34384362, PMID: 35965679, and PMID: 34537192), summarizing a paragraph AND highlighting the implications of all results in the paragraph.


------[Minor Concerns about writing]
1. Throughout the manuscript, it seems better to use Grammarly (https://www.grammarly.com/) to check & correct potential grammatical errors or typos. For example,
1.1 In Conclusions of ABSTRACT ("The results provide insights into MX1 and UBE2L6 as potential common metaflammation biomarkers and gene and pathway targets for DM and AS"), it seems better to change the sentence into "The results provide insights into MX1 and UBE2L6 as potential common metaflammation biomarkers and targets for DM and AS". After this revision, the sentence would be clearer and more concise.

2. In ABSTRACT:
2.1 In Results ("The top 10 hub proteins, namely, ISG15, IRG6, IRF7, IFIT3, MX1, UBE2L6, DDX58, IFIT2, USP18, and IFI44L, drive aspects of DM and AS"), it seems better to change this sentence into "The top 10 hub proteins, which were predicted to promote DM and AS, were ISG15, IRG6, IRF7, IFIT3, MX1, UBE2L6, DDX58, IFIT2, USP18, and IFI44L". After this revision, the sentence would be more rigorous, because the evidence could be not strong enough to conclude "drive".

3. In INTRODUCTION:
3.1 It would be clearer (easier to read) to split INTRODUCTION into two to three paragraphs and add one blank line after each paragraph (a pattern like PMID: 37452367, PMID: 34715879, PMID: 34384362, PMID: 35965679, and PMID: 34537192).

Reviewer 4 ·

Basic reporting

The manuscript is well organized and executed. I appreciate the authors contribution for this work that points out a significant objective. Overall the work seems to be robust though I have some concerns that are pointed out in below sections.

Experimental design

The experimental design seems to be robust but I would suggest to clarify and explain the difference in the groups used in the comparison -HC + STZ and STZ+HC more elaborately.

Validity of the findings

The findings of the study appear to be robust through a strong computational pipeline though the validation can not stand alone through computational analysis. It would be great if the authors can at least validate these markers to some extent through wet lab approach. If that is out of scope of the manuscript I appreciate that the authors have mentioned that further validation is required. Other than these points I would like to indicate the figure quality issue through fig 3-5. The legibility in these figures are not optimal and requires regeneration of these figures in high quality.

Figure 6 requires the explanation of color gradient for top 10 hub genes in figure legend.

I would also suggest to include an excel sheet for all 103 DEGs with details in the supplemental data.

---

## Round 0.2 · Minor Revisions

Although reviewers were satisfied by your revision, Section Editor made the following remarks:

This is an interesting study but I have 3 comments that could make this a better paper:

1. Why streptozotocin? That was not explained anywhere in the study. I looked it up and found that "Streptozotocin or streptozocin (INN, USP) (STZ) is a naturally occurring alkylating antineoplastic agent that is particularly toxic to the insulin-producing beta cells of the pancreas in mammals." That makes sense but for those (like me) not familiar the mode of action, I think this information would be good to add somewhere in the manuscript.

2. Also missing is critical information what genome assembly was used to map reads - I only find this:
From the genome website, we downloaded the annotation files for the reference genome and gene model directly." What reference genome and annotation files? I assume this was Sus scrofa? I found 33 genomes for S. scrofa at NCBI. So, we need to know which genome assembly and annotation files were used in this study.

3. Finally, since MX1 and UBE2L6 are the most crucial genes at the intersection of DM and AS, I think it should be explained what these genes are at first mention in the abstract and results, more so than what is in the discussion. I found that Myxoma resistance (Mx)1 is an interferon-stimulated gene that mediates potent anti-influenza A virus activity in mice. That is interesting from the standpoint of DM and AS - could there be a viral component? Ubiquitin/ISG15-conjugating enzyme E2L6 (UBE2L6) encodes a ubiquitin-modifying protein that targets them for degradation. How could that activity be involved in DM and AS?

Addressing these concerns could improve the manuscript for publication.

Reviewer 3 ·

Basic reporting

Please see Additional Comments.

Experimental design

Please see Additional Comments.

Validity of the findings

Please see Additional Comments.

Additional comments

Thank the authors for responding to all of the comments. The current version has been improved.

---

## Round 0.3 · accepted · Accept

All issues pointed out by the Section Editor were adequately addressed and the revised manuscript is acceptable now.